# Own Method of Forehead Skin-Sinus Fistula Treatment with Enabling and Maintaining Physiological Drainage in Overgrowing Inflammation of the Sinuses-A Case Report

Marek Łapok [1,*], Michał Polguj [2], Jarosław Miłoński [3] and Marcin Kozakiewicz [4,*]

1    Department of Angiology, Medical University of Lodz, 90-752 Lodz, Poland
2    Department of Normal and Clinical Anatomy, Medical University of Lodz, 90-752 Lodz, Poland
3    Department of Otolaryngology, Laryngological Oncology, Audiology and Phoniatrics,
     Medical University of Lodz, 90-549 Lodz, Poland
4    Department of Maxillofacial Surgery, Medical University of Lodz, 90-549 Lodz, Poland
*    Correspondence: marek.lapok@umed.lodz.pl (M.Ł.); marcin.kozakiewicz@umed.lodz.pl (M.K.)

**Abstract:** Surgical treatment of craniofacial region diseases associated with soft tissue and bone loss is always a challenge for medical specialists. The paper reports a case of a 70-year-old patient who presented with a defect in the forehead area that had been dermatologically treated. Following clinical, laboratory and imaging diagnostics, the lesion was classified as skin sinus fistula. The case study analyzes overgrowth of the nasofrontal duct associated with the laryngological endoscopic procedure (restoring the patency of the frontal sinus was performed as the initial part of the treatment.) Maxillofacial surgeons made an attempt to create an acrylic "space maintainer" which can be used either temporarily or as the ultimate option. The manuscript also describes a method of multi-specialized surgical treatment and subsequent esthetic management with the use of an individualized epithesis for soft tissue replacement.

**Keywords:** skin sinus fistula; free flaps; maintainer; intra-fistular approach; epithesis; case report

## 1. Introduction

Surgical methods applied in the craniofacial region should be minimally invasive and aim to preserve physiological function and esthetics. The unpaired frontal bone includes two triangular frontal sinuses. Under physiological conditions, they drain through the nasofrontal duct running into the region of the semilunar hiatus [1]. If patency is maintained, this connection provides heating and humidification of the air and ensures vocal resonance and mechanical protection of the cerebral region of the skull [2].

The normal passage of mucus secretions may be blocked for many reasons. The paranasal sinuses are often subject to chronic inflammation and the consequent release of pro-inflammatory cytokines results in tissue remodeling with fibrosis, atrophy or hypertrophy occurring in the basement membrane of the mucosa. The resulting mechanical obstructions impair the function of the sinus epithelium. [3,4] One of the rare complications of chronic sinusitis is a skin-sinus fistula [5].

Another potential etiological element is metastatic or dystrophic pathological calcification, which may develop in leukemia. [4] It increases the tendency for overgrowth of physiological sinus drainage pathways or the destruction of bone tissue.

The correlation between *Staphylococcus aureus* infection and osteomyelitis [5] should not be forgotten. If chronic, it may lead to osteonecrosis and consequently result in fistula formation.

Other key factors in the etiopathogenesis of the defect that should not be ignored are the internal disorders occurring in hematological diseases. These are increased tissue metabolism or cell apoptosis.

This manuscript aims to present a case of a patient with the defect resulting from skin sinus fistula and to describe a novel technique, i.e., "space maintainer ensuring physiological duct", which can be used either temporarily or as the ultimate method when more complicated procedures cannot be used due to the patient's general condition.

## 2. Case Study

A 70-year-old patient presented with a forehead defect, approximately 3 cm in diameter (Figures 1 and 2). He reported that the lesion around the glabella had developed about six months before first appointment at the Maxillo-Facial Clinic. At that time, due to the COVID-19 pandemic, he was wearing a visor, which pressed the forehead [6]. The lesion resembled a carbuncle [7] and the patient was referred to a dermatologist; however, his local condition had since deteriorated.

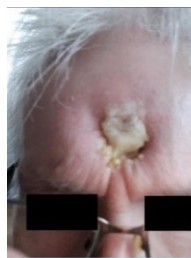

**Figure 1.** Initial condition.

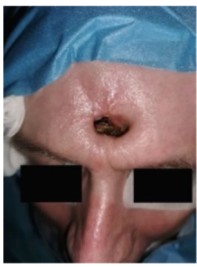

**Figure 2.** Condition pro the procedure.

He gave a history of myeloid leukemia diagnosed 6 years earlier. The disease had been regularly monitored and treated with Imatinib. A sample of secretion from the frontal sinus was positive for *Staphylococcus aureus*. An antibiogram test was performed to administer the right antibiotic. Histopathological examination showed chronic inflammation of the frontal sinus.

TIMELINE
1. Dermatological treatment.
2. Differential diagnosis (anamnesis, histopathological examination, CT scanning, Figures 1–3).
3. Plan of treatment No. 1 (laryngological surgery, Figures 5 and 6).
4. Overgrowth of the nasofrontal duct (CT scanning, Figure 3).
5. Plan of treatment No. 2 (neurosurgery with plastic surgery, Figures 7 and 8)–refused con-sent.
6. Final plan (maxillofacial surgery, "space maintainer", epithesis, Figures 9–11).

The patient underwent a non-contrast CT scan (Revolution Evo system, 64 rows, GEHealthcare). The obtained image revealed a defect of bone and soft tissue in the glabella area, about 3 cm in diameter, and an overgrowth of bone tissue across the entire width of the nasofrontal duct. (Figure 3A,B)

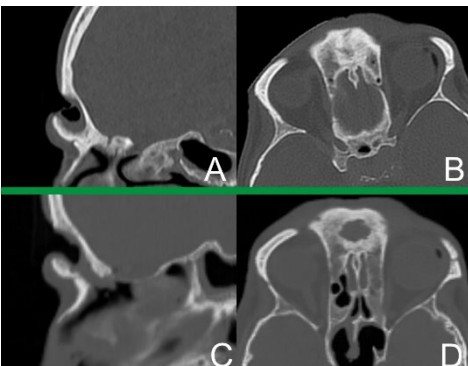

**Figure 3.** (**A**) Condition after the first CT scan-sagittal plane. (**B**) Condition after the first CT scan-transverse plane. (**C**) CT scan 14 days after first procedure-sagittal plane. (**D**) CT scan 14 days after first procedure-transverse plane.

At a medical case meeting, a team of laryngology and maxillofacial surgery specialists made the decision to apply a two-stage procedure. The first stage was aimed at achieving physiological drainage and ventilation of the sinus. The second stage was planned to be performed after approximately 10–14 days in order to reconstruct the cavity of the glabella.

The patient was presented with the following treatment options:

- Esser's double pedicle flap [8] Figure 4
- radial forearm flap [9]
- temporoparietal fascia free flap [10]

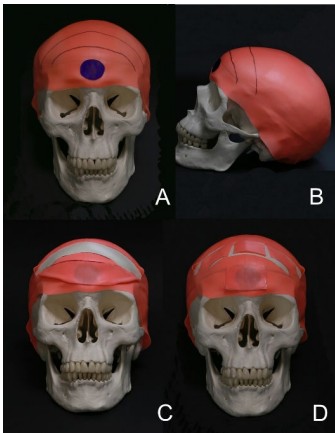

**Figure 4.** Esser's flap diagram on a plastic model of the skull. (**A**) Defect and the incision site-front view. (**B**) Defect and the incision site-lateral view. (**C**) Inserting the flap to replace the defect. (**D**) Final model.

During the procedure, the ethmoid bone was resected bilaterally and the hypertrophic mucosa was removed. Hypertrophic lesions were resected from the left and right frontal sinus drainage pathways and the hypertrophic mucosa was removed. Using a cutter, the bone margin of the anterior wall of the right and left frontal sinuses and the ethmoid bone were removed bilaterally, along with a fragment of the nasal septum. Two drains were inserted bilaterally through the anterior wall of the frontal defect and brought out through the nasal meatuses. Tampons were inserted anteriorly into both nasal meatuses (Figures 5 and 6).

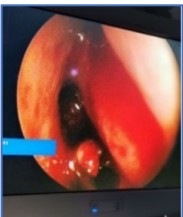

**Figure 5.** Condition before the endoscopy-necrotic tissues.

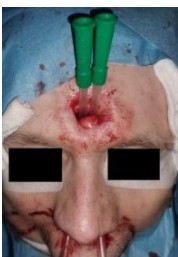

**Figure 6.** Condition after the laryngological procedure.

Fourteen days following the laryngological procedure, the created canal was found to be overgrown and the nasofrontal duct lacked patency. A repeat CT scan of the facial part of the cranium confirmed that the canal was overgrown with soft tissue compared with the previous examination (Figure 3C,D).

Another medical case meeting was held, this time with specialists in neurosurgery and maxillofacial surgery. Frontal sinus obliteration was planned: however, a defect was noted in the posterior wall of the frontal sinus (Figure 7) together with a 10 mm-wide fluid space in the temporoparietal region (Figure 8) with a density of 15 Hounsfield units, a dilated Sylvian fissure. Therefore frontal sinus cranialization was eventually proposed.

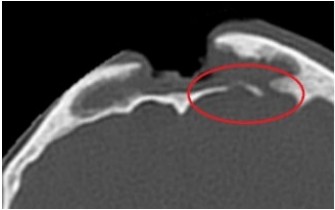

**Figure 7.** Defect in the posterior wall of the frontal sinus, CT scan.

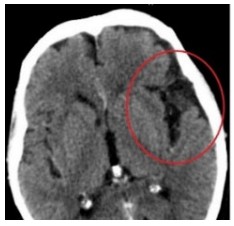

**Figure 8.** Fluid space in the frontotemporal region, CT scan.

However, the patient refused to undergo the procedures due to complications that developed during the initial stages and the uncertain prognosis.

A team of oral and maxillofacial surgeons created a 3D-printed acrylic "space maintainer" through segmentation of bone structures. (Figure 9A). In the next step, two maintainers were designed, one (No. 1) with a split end intended to fit the right and left nasal passages (Figure 9B,C) and another (No. 2) with a special "holder" to facilitate intraoperative adjustment (Figure 9D).

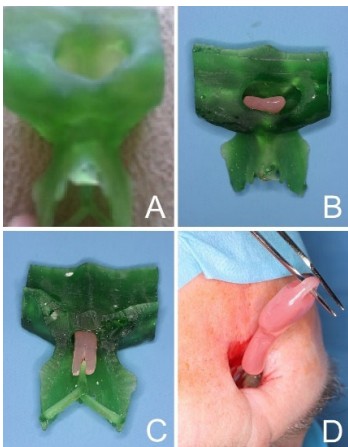

**Figure 9.** "Space maintainer" design (**A**) 3D model (**B**) Maintainer No. 1 (**C**) Maintainer No. 1 (**D**) Maintainer No. 2.

In the first stage of the procedure, any necrotic soft tissues in the upper layer were removed (Figure 10A). Plastic surgery of the osseous canal was performed (Figure 10B) using an intra-fistular approach. Maintainer No. 2 was used first to create the main canal and this was followed by Maintainer No. 1 to fit precisely into the nasal passages (Figure 10C). Maintainer No. 2 was left in the created canal (Figure 10D).

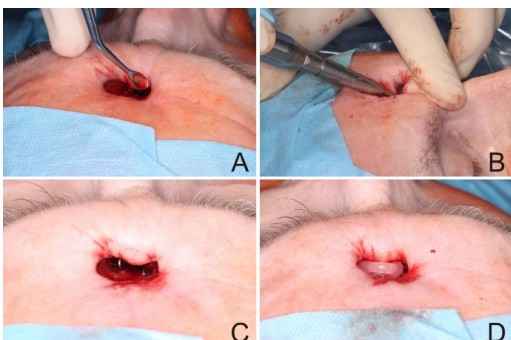

**Figure 10.** Maxillofacial procedure. (**A**) Removal of tissue from the frontal sinus. (**B**) Intra-fistular approach. (**C**) Unblocking the nasal passages. (**D**) Fitted maintainer.

An epithesis was then created to isolate the frontal sinus cavity and improve the appearance of the treated area. Prior to epithesis fixation, the holder was removed from maintainer No. 2 to increase the surface area (Figure 11).

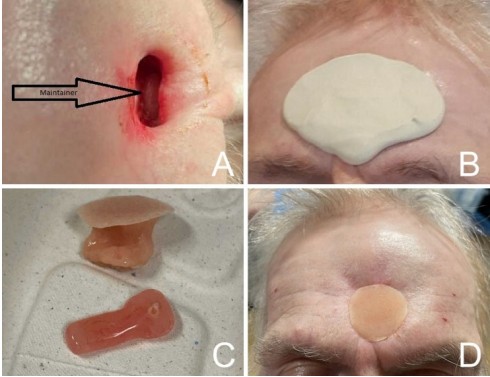

**Figure 11.** Prosthesis procedure. (**A**) Condition after correction of the maintainer. (**B**) Taking impression for an epithesis. (**C**) Epithesis and maintainer in the final version. (**D**) Epithesis inserted into the defect area.

## 3. Discussion

It should be emphasized that there are numerous factors that may have led to the development of this rare case, therefore several diseases are worth including in differential diagnosis aimed at identifying similarities between different conditions. One example is Pott's puffy tumor-subperiosteal abscess of the frontal bone with concomitant frontal sinusitis. Other nonobvious causes of Pott's puffy tumor have since been noted such as insect bites or the use of drugs by inhalation. It can also develop as a complication following acute mastoiditis, a dental infection, a neurosurgical procedure, an insect bite or use of drugs by inhalation. The condition most often affects young adults, but may be observed in any age group, occurring more frequently among men [11–14].

A similar presentation is shared by osteomyelitis, caused by an inflammatory process that can develop in any of the paranasal sinuses [5].

A paranasal sinus mucocele is a cystic structure filled with mucus and exfoliated epithelium [15–17]. It most often develops by forming an anatomical stenosis, then blocking the normal passage. It continuously grows and compresses adjacent tissues, resulting in potential bone defects.

If left untreated, diseases of the paranasal sinuses can lead to the spread of inflammation affecting adjacent tissues, especially in immunocompromised patients [18–20].

Such a lesion can occur intra-orbitally or even intracranially. The close presence of the meninges and the venous sinuses, especially the cavernous sinus, can cause threatening complications requiring neurosurgical treatment as they can even have fatal consequences [18–20].

During comprehensive treatment, the initial stage involves a commonly applied laryngological procedure called Draf III based on modern endoscopic techniques. The trans-nasal pathway leads to the frontal recess in the initial step. The next step involves complete removal of the inferior wall of the frontal sinus with partial resection of the anterior part of the perpendicular plate of the middle nasal concha [3]. In this clinical situation, due to the defect in the anterior wall of the frontal sinus, the intra-fistular approach was applied.

The first references to the use of pedicled flaps date back to the 19th century, when a pedicled flap based on the inferior epigastric vessels was used to reconstruct the forearm of a patient who had suffered a burn injury [8]. In addition, a stitched flap of the submandibular region was also used to reconstruct the buccal region after tumor removal [8]. Dunham describes a procedure for reconstructing a buccal defect with a unilateral pedicled flap based on the superficial temporal artery; in this case, the defect of the removed flap was covered with a free skin graft [8]. Finally, an island flap based on the temporal artery was used to reconstruct a defect of the lower eyelid by insertion into a tunnel created along the lateral edge of the orbit [8].

Esser described a visor flap with a narrow pedicle based bilaterally on the temporal arteries. The developed plastic models were prepared according to his procedure [8].

The first stage involves dissection of a bilateral pedicle flap on the superficial temporal vessels (Figure 4A,B).

The next step is to move the flap towards the defect area. A free skin graft is inserted in the space left by the dissected flap for about 10–14 days. The pedicle is then cut off and placed in the recipient site (Figure 4C,D).

Due to the overgrowth of the created canal during the laryngological procedure, neurosurgical treatment was attempted. The key objective was to remove the mucosa, which in the future would impair wound healing after Esser's procedure, from the frontal sinus. Two methods were presented: the first was frontal sinus obliteration which was first applied as early as in the 19th century. The procedure involves external access and total removal of the sinus along with the periosteum. The next step is the insertion of obliteration material such as gelatin, blood clot, bone, muscle, polymethacrylate, or hydroxyapatite [21]. However, autogenous adipose tissue proved to be the best option as it quickly undergoes revascularization and connective-tissue remodeling, as well as partial resorption [21]. Next,

it was planned to cover the defect with a visor flap. One of the absolute contraindications is potential intracranial complication, and in this case it was very likely to occur due to the bony defects in the posterior wall of the frontal sinus. It would be necessary to perform cranialization of the frontal sinus, involving a complete removal of the posterior bony wall of the sinus, usually with the need to supply the damaged dura mater. [22]

As a result of advances in microsurgery, free flaps have been popularized. The present case discusses the treatment options with the use a radial free flap or a temporoparietal fascia free flap. Allen's test was negative bilaterally. We informed the patient that the prognosis was very uncertain due to leukemia, the size of the defect and *Staphyllococcus aureus* infection. The scar that could potentially appear on the forearm was one of the reasons why the patient refused to undergo the treatment.

In the analyzed clinical condition, an attempt was made to maintain the space of the nasofrontal canal with a prosthetic material that provides patency and ventilation of the frontal sinus.

The final step was epithesis insertion. This procedure is often performed when plastic surgery methods fail or the patient does not consent to the proposed treatment, as was the case here. It is primary function is to generally improve the esthetic appearance [23].

The epithesis should cover the prosthetic area with a margin of shaded material and reach into the hollows of folds and wrinkles, possibly in the immobile area. To ensure retention of a facial epithesis, glass is commonly inserted into a facial epithesis; however, metal implants can also be used to fix the ear or nose epithesis [23]. Modern material science makes it possible to create a work that restores individual anatomical features and ensures proper esthetics [24]. Many tests in the field of mechanics and biomechanics have been performed to determine the appropriate tear strength. It most often occurs during removal, as they are made and placed with the help of a retention form. In a trouser-tear test, a thin layer of elastomer is made in the shape of trousers. The legs of the "trousers" are then pulled away from each other and the energy required to expand the tear is measured.

## 4. Conclusions

Unfortunately, no common thoroughly-described treatment method currently exists for skin-sinus fistulae. Each case must be planned on an individual basis and the treatment strategy selected according to the specific case and accompanying factors. Furthermore combined full-thickness dermo-osseal defects are a problematic issue for clinicians around the world.

The novelty of our case report indicates that the "space maintainer" can be used either temporarily or as an ultimate solution. We attempted to use the "space maintainer" temporarily to create a via nasofrontalis before potential plastic surgery. In the case of patients who are not qualified for more complicated procedures due to their general health condition, ecto-prothesis is used as the ultimate option.

Our patient has been regularly assessed at the outpatient clinic in order to correct the maintainer.

He has not reported any complaints and his quality of life has improved considerably.

**Author Contributions:** Conceptualization M.Ł., M.P. and M.K.; methodology, M.Ł., J.M. and M.K.; validation, M.Ł., M.P., J.M. and M.K.; formal analysis, M.P. and M.K.; resources, M.Ł., M.P. and M.K.; data curation, M.Ł.; writing—original draft preparation, M.Ł.; writing—review and editing, M.Ł., M.P., J.M. and M.K.; visualization, M.Ł.; supervision, M.P. and M.K.; funding acquisition, M.P. and M.K. All authors have read and agreed to the published version of the manuscript.

**Funding:** This research was funded by the Medical University of Lodz 503/1-031-01/503-11-001-19-00 and 503/1-138-01/503-51-001-19-00.

**Institutional Review Board Statement:** Not applicable.

**Informed Consent Statement:** Written informed consent has been obtained from the patient to publish this paper.

**Data Availability Statement:** Not applicable.

**Conflicts of Interest:** The authors declare no conflict of interest.

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
