# Peer review of "Own Method of Forehead Skin-Sinus Fistula Treatment with Enabling and Maintaining Physiological Drainage in Overgrowing Inflammation of the Sinuses-A Case Report"

_applsci, doi:10.3390/app13042107_

Round 1
Reviewer 1 Report (New Reviewer)
The case report you submitted is very poor in quality. I hope that it will be expressed more intensively and unnecessary things will not be listed.
1. Indicate that it is a case report in the title.
2. Concentration on narrative is reduced. Write more concisely.
3. Delete "Table 1" and change it to narrative instead. You only need to use antibiotics that have sensitivity to bacteria.
4. Delete "Table 2" and briefly describe it. Since CAD-CAM technology is commonly applied to various parts of the human body, there is nothing special about it.
5. How long is the follow-up period after the "epithesis" is delivered? Have there been any recurrences or complications of the disease during that period?
6.As an "Applied Sciences" SCI journal, it is read by many researchers around the world. Therefore, it is recommended that the selected reference be written in English, preferably a SCI/SCIE journal. A significant revision of the reference is required. Replace #2, #3, #4, #8, #18, #19, #20, #21, #23, #25, and #29 references.
Author Response
Dear Reviewer 1
We would like to thank Reviewers for taking the necessary time and effort to review our work. We sincerely appreciate all your valuable comments and suggestions, which helped us in improving the quality of the text. We considered your comments and made the requested corrections.
We are very grateful for giving us the opportunity to publish the article in Advances in Maxillofacial and Oral Surgery
Yours sincerely,
Marek Łapok
1.Indicate that it is a case report in the title.
We changed the title.
2.Concentration on narrative is reduced. Write more concisely.
We reduced the manuscript as suggested.
3.Delete "Table 1" and change it to narrative instead. You only need to use antibiotics that have sensitivity to bacteria.
We deleted “Table 1” and changed it to narrative instead.
4.Delete "Table 2" and briefly describe it. Since CAD-CAM technology is commonly applied to various parts of the human body, there is nothing special about it.
We deleted “Table 2” and briefly described it instead.
5.How long is the follow-up period after the "epithesis" is delivered? Have there been any recurrences or complications of the disease during that period?
I saw the patient for the first time in July 2021. He said that the defect occurred in February 2021. He began to be treated dermatologically then. Later, we implemented the subsequent stages of our treatment, i.e. CT (July, 2021), laryngological surgery (September,2021), another CT (November, 2021), maxillofacial surgery (December, 2021), epithesis (January, 2022). This patient was regularly assessed at the outpatient clinic in order to correct the maintainer. The patient has not reported any complaints and his quality of life has improved.
6.As an "Applied Sciences" SCI journal, it is read by many researchers around the world. Therefore, it is recommended that the selected reference be written in English, preferably a SCI/SCIE journal. A significant revision of the reference is required. Replace #2, #3, #4, #8, #18, #19, #20, #21, #23, #25, and #29 references."
We changed the references.

Reviewer 2 Report (New Reviewer)
Please correct the Introduction in lines 31/32
In discussion- Are there any other ways that was possible to treat the patience? discuss about the it in one paragraph
Author Response
Dear Reviewer 2
We would like to thank Reviewers for taking the necessary time and effort to review our work. We sincerely appreciate all your valuable comments and suggestions, which helped us in improving the quality of the text. We considered your comments and made the requested corrections.
We are very grateful for giving us the opportunity to publish the article in Advances in Maxillofacial and Oral Surgery
Yours sincerely,
Marek Łapok

Reviewer 3 Report (New Reviewer)
Dear Editor and Authors,
Thank you for the opportunity to review the manuscript entitled “Own method of forehead skin-sinus fistula treatment with enabling and maintaining physiological sinus drainage in over-growing inflammation of the estuary”. The authors aimed to report a case of a 70-year-old patient who was treated due to skin sinus fistula. The case describes a method of multispecialty surgical treatment and subsequent esthetic management with the use of an individualized epithesis for soft tissue replacement. The study is interesting and should be of interest of the Journal readers. However, I have some remarks:
- Abstract: I suggest expanding the abstract and adding more information about the used technique and explaining “after Draf III” – this is not clear for unfamiliar readers
- Introduction: “One of rare complications is a skin-sinus fistula” – complication of what? Please provide more theoretical information about these fistulas – aetiology (what about postoperative cases?), symptoms, possible treatment, and then at the end specify the aim of your report highlighting the novelty of your report. (the aim is to present a case of a patient with ….. and describe a novel and…. technique….which can act as alternative to a classic….etc….)
- Case report: “At that time, he was wearing a visor due to the COVID-19 pandemic” – was this a reason of the fistula???? What is the logic in reference 6 and 7 in this location? What was the reason of the fistula? Have you looked for similar cases in the literature?
- “He gave a history of myeloid leukemia treated with Imatinib.” Was he during the treatment or had it been already completed years ago?
- I am not sure if table 1 is needed
- “Esser’s double pedicle flap” – a figure showing the idea of the flap would be helpful
- “After 14 days following the laryngological procedure, overgrowth of the created canal and lack of physiological patency of the nasofrontal duct.” – I do not understand the sentence
- “A team of oral and maxillofacial surgeons made an attempt (table 2)” – where is table 2? Should be closer to the text and its description in the text should be more informative
- Discussion: “Several pathological diseases are worth discussing in detail for differential diagnosis. Also, some similarities should be identified.” – what this sentence refers to? What are “pathological diseases” ?
- Please discuss further proceedings in your patient. Is this epithesis going to be a definite treatment in your patient? If you suggest it to be definite provide limitations and for whom such treatment can “work” but also that this can be a kind of temporary solution before final treatment…
- “The first references to the use of pedicled flaps date back to the 19th century. Wood of Yorkshire described a case of a pedicled flap based on the inferior epigastric vessels reconstructing the….” I think this historical paragraph adds nothing important, I would focus on discussing modern techniques and technologies of designing episthetics
- “In developmental dentistry, space maintainers are used to avoid dental crowding” – I do not know what this has to do with the case
- Conclusion – should be rephrased to highlight the novelty of the case /was such treatment previously described in the literature? /, and it should be clearly stated that the technique can be useful in a specific group of patients /in whom?/
- Needs proof-reading /English editing/!
Author Response
Dear Reviewer 3
We would like to thank Reviewers for taking the necessary time and effort to review our work. We sincerely appreciate all your valuable comments and suggestions, which helped us in improving the quality of the text. We considered your comments and made the requested corrections.
We are very grateful for giving us the opportunity to publish the article in Advances in Maxillofacial and Oral Surgery
Yours sincerely,
Marek Łapok
1. Abstract: I suggest expanding the abstract and adding more information about the used technique and explaining “after Draf III” – this is not clear for unfamiliar readers
We introduced changes in lines 13-15.
2. Introduction: “One of rare complications is a skin-sinus fistula” – complication of what? Please provide more theoretical information about these fistulas – aetiology (what about postoperative cases?), symptoms, possible treatment, and then at the end specify the aim of your report highlighting the novelty of your report. (the aim is to present a case of a patient with ….. and describe a novel and…. technique….which can act as alternative to a classic….etc….)
We introduced changes in lines 27-46 and 139-140.
3. Case report: “At that time, he was wearing a visor due to the COVID-19 pandemic” – was this a reason of the fistula???? What was the reason of the fistula? Have you looked for similar cases in the literature?
We are not entirely sure what caused the defect. We considered all the pathologies that occurred in this case – leukemia, osteomyelitis, Staphylococcus aureus infection, sinusitis, anatomical anomalies,wearing visor due to COVID 19. I was not able to find similar case studies, therefore I could not make any comparisons.
4.What is the logic in reference 6 and 7 in this location?
The patient first observed the defect that occurred in the glabella area during the COVID-19 pandemic, when for a few weeks he was wearing a visor that pressed his forehead. He was referred to a dermatologist and diagnosed with a furuncle-like lesion . Since the patient’s local condition had deteriorated, he presented to the Maxillofacial Surgery Clinic.
We introduces changes in lines 50-52.
5.“He gave a history of myeloid leukemia treated with Imatinib.” Was he during the treatment or had it been already completed years ago?
We introduced changes in lines 53-54.
6.I am not sure if table 1 is needed
We deleted Table 1.
7. “Esser’s double pedicle flap” – a figure showing the idea of the flap would be helpful
We introduced changes in lines 157-169.
8.“After 14 days following the laryngological procedure, overgrowth of the created canal and lack of physiological patency of the nasofrontal duct.” – I do not understand the sentence
We changed the sentence,lines 88-89.
9. “A team of oral and maxillofacial surgeons made an attempt (table 2)” – where is table 2? Should be closer to the text and its description in the text should be more informative
We deleted “Table 2” and described it instead.
10. Discussion: “Several pathological diseases are worth discussing in detail for differential diagnosis. Also, some similarities should be identified.” – what this sentence refers to? What are “pathological diseases” ?
We introduced changes in lines 131-132.
11. Please discuss further proceedings in your patient. Is this epithesis going to be a definite treatment in your patient? If you suggest it to be definite provide limitations and for whom such treatment can “work” but also that this can be a kind of temporary solution before final treatment…
We explained it in lines 43-46.
12.“The first references to the use of pedicled flaps date back to the 19th century. Wood of Yorkshire described a case of a pedicled flap based on the inferior epigastric vessels reconstructing the….” I think this historical paragraph adds nothing important, I would focus on discussing modern techniques and technologies of designing episthetics
We deleted these lines.
13.“In developmental dentistry, space maintainers are used to avoid dental crowding” – I do not know what this has to do with the case
We deleted these lines.
14.Conclusion – should be rephrased to highlight the novelty of the case /was such treatment previously described in the literature? /, and it should be clearly stated that the technique can be useful in a specific group of patients /in whom?/
We referred to it in lines 201-204.

Reviewer 4 Report (New Reviewer)
This paper reports a method of forehead skin-sinus fistula treatment with enabling and maintaining physiological sinus drainage in overgrowing inflammation of the estuary based on one clinical case. Overall, it is an extremely hard to read paper, there is no clear argument line to follow. Proof-reading service is mandatory before to complete the peer-review.
Author Response
Dear Reviewer 4
We would like to thank Reviewers for taking the necessary time and effort to review our work. We sincerely appreciate all your valuable comments and suggestions, which helped us in improving the quality of the text. We considered your comments and made the requested corrections.
We are very grateful for giving us the opportunity to publish the article in Advances in Maxillofacial and Oral Surgery
Yours sincerely,
Marek Łapok

Round 2
Reviewer 1 Report (New Reviewer)
This manuscript has been well revised and I agree to publish it.
Author Response
Dear Reviewer 1
Thank you for reviewing our manuscript an interest of our work.

Reviewer 3 Report (New Reviewer)
Dear Editor and Authors,
Thank you for the opportunity to review a revision of the manuscript entitled “Own method of forehead skin-sinus fistula treatment with enabling and maintaining physiological sinus drainage in over-growing inflammation of the estuary”. The study is interesting and should be of interest of the Journal readers. The authors made some corrections according to the suggestions, which made their case report clear and more accessible for the Readers. However, I still have some remarks:
- Abstract: Still “Draf III” is not explained and is not clear for unfamiliar readers.
- In the aim you stated: “patient with atypical nature of the defect” – I am not sure if “atypical nature of the defect” is right here/?/. I think “the defect” should be specified here, did you mean “unknown etiology” saying “atypical nature”?
- Case report: Again, the idea of “inverted Draf III” should be explained.
- Still, some proof-reading is needed.
Author Response
Dear Reviewer 2
Thank you for reviewing our manuscript an interest of our work. We are very grateful for the review
remarks because they helped us to improve the paper. All Reviewer’s remarks
were applied to the new version of the manuscript and requested corrections were made. We are very grateful for opportunity to publish the article in Advances in Maxillofacial and Oral Surgery
Your sincerely Marek Łapok
1.Abstract: Still “Draf III” is not explained and is not clear for unfamiliar readers.
We deleted the term “Draf III” and used “laryngological endoscopic procedure” instead. The later expression should be clearer to the readers. Lines 159-160 describe Draf III, so it is explained.
2.In the aim you stated: “patient with atypical nature of the defect” – I am not sure if “atypical nature of the defect” is right here/?/. I think “the defect” should be specified here, did you mean “unknown etiology” saying “atypical nature”?
We deleted the word “atypical” because it is redundant and makes the manuscript more complicated.
3.Case report: Again, the idea of “inverted Draf III” should be explained.
We changed the expression. The term “intrafistular approach” is used in the revised version.
- Still, some proof-reading is needed.

Reviewer 4 Report (New Reviewer)
Despite many corrections made for this version, it is really hard to follow what is relevance of this case. I suggest to:
- Use the CARE statement to structure the manuscript. https://www.care-statement.org/checklist
- Delete all the short sentences introduced across the paper. Authors must to assure a clean argument line. For example, what is intendend with the sentence at line 27-28? 131-132? 139-140?
- It will be helpful to construct composite figures to shown in a solo figure what happen at any stage of the case.
- Discussion must explicitly refer to your case, it is not adequate to include info about pathologies that are not clearly and explicitly linked to your case.
- Conclusion must address what is the relevance of this new method used to treat this case.
Author Response
Dear Reviewer 4
Thank you for reviewing our manuscript an interest of our work. We are very grateful for the review
remarks because they helped us to improve the paper. All Reviewer’s remarks
were applied to the new version of the manuscript and requested corrections were made. We are very grateful for opportunity to publish the article in Advances in Maxillofacial and Oral Surgery
Your sincerely Marek Łapok
1.Use the CARE statement to structure the manuscript. https://www.care-statement.org/checklist
A.Key words-we changed them.
B.Timeline-we included it.
C.The primary “take-away” lessons from this case report (without references) are presented in a one paragraph conclusion - lines 206-210.
D.Patient Perspective -lines 211-212.
2.Delete all the short sentences introduced across the paper. Authors must to assure a clean argument line. For example, what is intendend with the sentence at line 27-28? 131-132? 139-140?
We deleted the lines.
3.It will be helpful to construct composite figures to shown in a solo figure what happen at any stage of the case.
We changed the figures. They should now be clearer to the readers.
4.Discussion must explicitly refer to your case, it is not adequate to include info about pathologies that are not clearly and explicitly linked to your case.
We are not entirely sure what caused the defect. We considered all the pathologies that occurred in this case – leukemia, osteomyelitis, Staphylococcus aureus infection, sinusitis, anatomical anomalies, wearing visor due to the COVID 19 pandemic and we wrote about the most important ones in discussion. We deleted short sentences in discussion in order to reduce the manuscript and present stronger arguments.
5.Conclusion must address what is the relevance of this new method used to treat this case.
We changed it.

Round 3
Reviewer 4 Report (New Reviewer)
This new version of the manuscript had improved my previous requests. I strongly suggest to proofread it by a language editing service before the publication.
This manuscript is a resubmission of an earlier submission. The following is a list of the peer review reports and author responses from that submission.
Round 1
Reviewer 1 Report
1. Case Study: For a patient with a complex pathology and clinical history I think is necessary to clarify when he was treated for leukemia, but more important when the frontal sinus fistula appears (months or maybe years before the patients was take in charge from the unit), if the fistula increased in the years or how many procedure the patients received.
2. Case Study: For the image 4a & 4b you have to choose a better picture!
3. Case Study: I believe it is appropriate to better explain the extent and management of the bone discontinuity of the posterior wall of the frontal sinus
4. Discussions: In the publication Walkden A, Tan N. Frontal Sino-Cutaneous Fistula Masquerading as a Basal Cell Carcinoma. Ear, Nose & Throat Journal. 2022;0(0). doi:10.1177/01455613221075235 The authors affirmed: "Our case demonstrates that by simply re-establishing the frontal sinus drainage pathway through a wide surgical neo-ostium, the fistula tract can heal spontaneously with excellent cosmetic results and prevent recurrent mucocele and fistula reformation. The Draf 3/MELP should be considered the gold standard procedure for complex pathology of the frontal sinus." Why you didn't think to wait if the fistula shows a regression?
5. Why after a possible regression you didn't try to make a skin pedicle flap or free flap to cover the skin fistula? There is some flaps like the temporoparietal fascia free flap used in head and neck (Molteni G, Gazzini L, Sacchetto A, Nocini R, Marchioni D. Role of the temporoparietal fascia free flap in salvage total laryngectomy. Head Neck. 2021 May;43(5):1692-1694. doi: 10.1002/hed.26602. Epub 2021 Jan 12. PMID: 33433928.) that it could be used also in this situation maybe maintaining his pedicle. Or also the radial free flap. I think you could improve the discussion of the possibility of reconstruction for this kind of skin defects with pedicle/free flaps.
Author Response
Dear Reviewer,
Thank You for reviewing our manuscript. We are very grateful for the remarks because they helped us to improve the paper. All Reviewer’s remarks were applied to the new version of the manuscript.We are very grateful for opportunity to publish the article in Advances in Maxillofacial and Oral Surgery
Your sincerely, Marek Łapok
1.Case Study: For a patient with a complex pathology and clinical history I think is necessary to clarify when he was treated for leukemia, but more important when the frontal sinus fistula appears (months or maybe years before the patients was take in charge from the unit), if the fistula increased in the years or how many procedure the patients received.
I saw the patient for the first time in July 2021. He said that the defect first occurred in February 2021. He began dermatological treatment then. Later, we implemented the subsequent stages of our treatment, i.e. CT (July, 2021), laryngological surgery (September,2021), another CT (November, 2021), maxillofacial surgery (December, 2021), epithesis (January, 2022). This patient was regularly assessed at the outpatient clinic in order to correct the maintainer.
He has been treated for leukemia for ten years.
2.Case Study: For the image 4a & 4b you have to choose a better picture!
According to Your’s suggestions, I choose a better picture.
3.Case Study: I believe it is appropriate to better explain the extent and management of the bone discontinuity of the posterior wall of the frontal sinus
In our discussion, we mention obliteration and cranialization as potential options. However, we did not perform those surgeries because the patient refused to give his consent being concerned about possible surgical complications.
4.Discussions: In the publication Walkden A, Tan N. Frontal Sino-Cutaneous Fistula Masquerading as a Basal Cell Carcinoma. Ear, Nose & Throat Journal. 2022;0(0). doi:10.1177/01455613221075235 The authors affirmed: "Our case demonstrates that by simply re-establishing the frontal sinus drainage pathway through a wide surgical neo-ostium, the fistula tract can heal spontaneously with excellent cosmetic results and prevent recurrent mucocele and fistula reformation. The Draf 3/MELP should be considered the gold standard procedure for complex pathology of the frontal sinus." Why you didn't think to wait if the fistula shows a regression?
In the first phase, the patient underwent endoscopic treatment.
After a few days, it was found that the canal after DrafIII had been overgrown, which was quite unexpected.
Following another CT scan, we proceeded to the next stage of the planned therapy involving the use of the maintainer with epithesis.
5.Why after a possible regression you didn't try to make a skin pedicle flap or free flap to cover the skin fistula? There is some flaps like the temporoparietal fascia free flap used in head and neck (Molteni G, Gazzini L, Sacchetto A, Nocini R, Marchioni D. Role of the temporoparietal fascia free flap in salvage total laryngectomy. Head Neck. 2021 May;43(5):1692-1694. doi: 10.1002/hed.26602. Epub 2021 Jan 12. PMID: 33433928.) that it could be used also in this situation maybe maintaining his pedicle. Or also the radial free flap. I think you could improve the discussion of the possibility of reconstruction for this kind of skin defects with pedicle/free flaps.
We talked to the patient about reconstruction with an Esser’s flap or a radial free flap, however, we informed him about a very uncertain prognosis considering leukemia, infection Staphylococcus aureus. The patient refused to give his consent. In the discussion section, we wrote about “INITIAL METHOD = DRAF III+ESSER’S RECONSTRUCTIVE SURGERY”.

Reviewer 2 Report
1) To provide more details info at the abstract. The abstract should provide background info about the patient , even before readers read the subsequent section of the manuscript
2) history of the patient is too short. whats the presenting complain? any previous management etc?
3) please insert ethical number
4) any short term and long term follow up? what the patient outcome at the end?
5) Discussion is ppor.. Why discuss about Osteomyelitis ? how does it related to the case? the history of the procedure is least important to be discussed. any comparison with similar case?
6) should highlight the main finding of this case and how can rthe reader benefits from it.
Author Response
Dear Reviewer,
Thank You for reviewing our manuscript. We are very grateful for the remarks because they helped us to improve the paper. All Reviewer’s remarks were applied to the new version of the manuscript.We are very grateful for opportunity to publish the article in Advances in Maxillofacial and Oral Surgery
Your sincerely,
Marek Łapok
1) To provide more details info at the abstract. The abstract should provide background info about the patient , even before readers read the subsequent section of the manuscript
Thank You for suggestion, we improved our abstract.
2) history of the patient is too short. whats the presenting complain? any previous management etc?
I saw the patient for the first time in July 2021. He said that the defect first occurred in February 2021. He began dermatological treatment. Later, we implemented the subsequent stages of our treatment, i.e. CT (July, 2021), laryngological surgery (September,2021), another CT (November, 2021), maxillofacial surgery (December, 2021), epithesis (January, 2022). This patient was regularly assessed at the outpatient clinic in order to correct the maintainer.
3) please insert ethical number
Informed Consent Statement: Any research article describing a study involving humans should contain this statement. Please add “Informed consent was obtained from all the subjects involved in the study.” OR “Patient consent was waived due to REASON (please provide a detailed justification).” OR “Not applicable.” for studies not involving humans. You might also choose to exclude this statement if the study did not involve humans.
Written informed consent for publication must be obtained from participating patients who can be identified (including by the patients themselves). Please state “Written informed consent has been obtained from the patient(s) to publish this paper” if applicable.
I sent permission to assistant editor in polish and englisch version,because here i can attached only one document
4) any short term and long term follow up? what the patient outcome at the end?
At the moment we are preparing a new, more esthetic ectoprosthesis. However, it is uncertain whether this attempt will be successful. The physiology was reconstructed, the patient’s comfort increased considerably.
5) Discussion is ppor.. Why discuss about Osteomyelitis ? how does it related to the case? the history of the procedure is least important to be discussed. any comparison with similar case?
We are not entirely sure what caused the defect. I considered all the pathologies that occurred in this case – leukemia, osteomyelitis, Staphylococcus aureus infection, sinusitis, anatomical anomalies. I was not able to find similar case studies, therefore I could not make any comparisons.
6) should highlight the main finding of this case and how can rthe reader benefits from it.
The main conclusion of our article is that it is necessary to consider possibly the least invasive methods that actually help our patients.
In the time of microsurgeries/free flaps, patients are not always willing to undergo such difficult surgeries. Flaps are not always successful, which in the future may also affect the patient’s general and local condition as well as his mental state.
In this case, sinus patency was regularly checked in the patient and it was found that the physiological state was restored.
We suggested the option of reconstruction with the use of an Esser’s flap, a radial free flap to the patient. We informed him about the uncertain prognosis due to his general condition, the size of the defect, Staphylococcus aureus infection. We also assured him we did not intend to treat his case as a medical experiment. The main purpose was to restore the function and esthetics.

Round 2
Reviewer 2 Report
The authos did not addressed most of the reviewers comment and not much of new information has been added to add the value to the manusscirpt. Suggest you to improve the flow of the mansscipt and add the suggestions as stated and send the manuscript for english proofing.